# Expression of Genes Encoding Selected Orexigenic and Anorexigenic Peptides and Their Receptors in the Organs of the Gastrointestinal Tract of Calves and Adult Domestic Cattle (*Bos taurus taurus*)

**DOI:** 10.3390/ijms25010533

**Published:** 2023-12-31

**Authors:** Katarzyna Kras, Katarzyna Ropka-Molik, Siemowit Muszyński, Marcin B. Arciszewski

**Affiliations:** 1Department of Animal Anatomy and Histology, Faculty of Veterinary Medicine, University of Life Sciences in Lublin, Akademicka 12 St., 20-950 Lublin, Poland; mb.arciszewski@wp.pl; 2Department of Animal Molecular Biology, National Research Institute of Animal Production, Krakowska 1 St., 32-083 Balice, Poland; katarzyna.ropka@iz.edu.pl; 3Department of Biophysics, Faculty of Environmental Biology, University of Life Sciences in Lublin, 13 Akademicka St., 20-950 Lublin, Poland; siemowit.muszynski@up.lublin.pl

**Keywords:** food intake, gastrointestinal tract, orexigenic peptides, anorexigenic peptides, cattle, real-time PCR

## Abstract

The regulation of food intake occurs at multiple levels, and two of the components of this process are orexigenic and anorexigenic peptides, which stimulate or inhibit appetite, respectively. The study of the function of these compounds in domestic cattle is essential for production efficiency, animal welfare, and health, as well as for economic benefits, environmental protection, and the contribution to a better understanding of physiological aspects that can be applied to other species. In this study, the real-time PCR method was utilized to determine the expression levels of *GHRL*, *GHSR*, *SMIM20*, *GPR173*, *LEP*, *LEPR*, and *NUCB2* (which encode ghrelin, its receptor, phoenixin-14, its receptor, leptin, its receptor, and nesfatin-1, respectively) in the gastrointestinal tract (GIT) of Polish Holstein–Friesian breed cattle. In all analyzed GIT segments, mRNA for all the genes was present in both age groups, confirming their significance in these tissues. Gene expression levels varied distinctly across different GIT segments and between young and mature subjects. The differences between calves and adults were particularly pronounced in areas such as the forestomachs, ileum, and jejunum, indicating potential changes in peptides regulating food intake based on the developmental phase. In mature individuals, the forestomachs predominantly displayed an increase in *GHRL* expression, while the intestines had elevated levels of *GHSR*, *GPR173*, *LEP*, and *NUCB2*. In contrast, the forestomachs in calves showed upregulated expressions of *LEP*, *LEPR*, and *NUCB2*, highlighting the potential importance of peptides from these genes in bovine forestomach development.

## 1. Introduction

The regulation of food intake in mammals occurs through intricate processes encompassing multiple levels, intertwined with a cyclic pattern of hunger and satiety referred to as the food intake cycle [1]. The gastrointestinal tract (GIT) and the nervous systems, including the central nervous system (CNS) and the enteric nervous system (ENS), engage in a two-way communication facilitated by the parasympathetic and sympathetic inputs, both through efferent and afferent fibers [2,3]. The efferent neuronal pathways play a crucial role in regulating the activity of the gut during both interdigestive and digestive phases [2,4,5]. In contrast, the afferent pathway transmits signals from various sensors in the intestine to the CNS. These sensors are sensitive to mechanical and chemical stimuli, including hormones, nutrients, and peptides [2,6]. There are numerous endogenous peptides in the GIT that act as regulators of food intake. Depending on their specific actions, they can be classified as either orexigenic peptides, which stimulate appetite, or anorexigenic peptides, which suppress appetite. The best-known orexigenic peptide in the GIT is ghrelin, which stimulates hunger sensations and promotes food intake [7]. In contrast, the anorexigenic adipokine leptin acts locally to decrease appetite [8]. The intricate interplay between anorexigenic and orexigenic peptides is essential for maintaining appropriate energy balance, body weight, and overall nutritional status [3,9,10].

The roles and mechanisms of action of some of the aforementioned well-known peptides are fairly well understood. However, new substances with molecular mechanisms that are not fully explained are constantly being discovered. Newly discovered appetite-regulating peptides, nesfatin-1 and phoenixin-14, were found in the hypothalamus and coexist in the same neuronal population [11]. As the hypothalamus acts as a central hub for the integration of signals that determine food intake, the function of peptides discovered in this structure, particularly in areas responsible for the regulation of food intake, is relatively well understood. However, the precise peripheral functions of these peptides in the GIT organs, which are the first to come into contact with ingested food, are not yet fully understood. It is also unclear to what extent these peptides are present in the GIT of different animal species.

The lack of literature on nesfatin-1 and phoenixin-14 in the GIT of domestic cattle (*Bos taurus taurus*), crucial contributors to global economies through meat and dairy production [12], warrants investigation. While existing studies focus on leptin, leptin receptor, and ghrelin, key elements like the ghrelin receptor, phoenixin-14, and nesfatin-1 remain unexplored [13]. This study aims to bridge these gaps, analyzing mRNA expression of orexigenic peptides and their receptors (*GHRL*, *GHSR*, *SMIM20*, *GPR173*) and anorexigenic peptides and their receptors (*LEP*, *LEPR*, *NUCB2*) in the GIT of domestic cattle.

Considering the complex stomach subdivision (rumen, reticulum, omasum, abomasum) in ruminant cattle, this research also delves into GIT segment-related variations. Additionally, calves undergoing intensive growth, weight gain, and a transition from pre-ruminants to ruminants, offer a unique perspective. This transformation may influence the physiology and functions of the GIT, even in animals with relatively developed forestomachs such as the calves included in this study [14].

Therefore, this study aims to investigate the mRNA expression patterns of key orexigenic and anorexigenic peptides, including ghrelin, leptin, and newly discovered peptides such as nesfatin-1 and phoenixin-14, and their receptors, across different segments of the gastrointestinal tract in domestic cattle, with a focus on age-related differences to enhance our understanding of the complex feeding mechanisms in polygastric animals and their implications for livestock health.

Understanding mRNA expression in the GIT of polygastric animals, distinct from their monogastric counterparts, provides valuable insights into intricate feeding mechanisms. Imbalances in these peptides expression might disrupt appetite regulation and nutrient utilization, potentially impacting the health and growth performance of cattle. Ultimately, this research contributes essential knowledge to enhance livestock well-being and optimize agricultural practices.

## 2. Results

### 2.1. mRNA Expression Levels

The real-time PCR analysis revealed the presence of mRNA of all genes in all examined segments in both age groups (Figure 1). The expression levels of *GHRL* varied significantly among the segments, with the highest mRNA levels in both groups observed in the abomasum, significantly lower levels in the duodenum, and negligible expression in the remaining GIT sections (*p* < 0.01) (Figure 1a).

In adults, the highest expression level of *GHSR* was observed in the ileum, slightly lower (but insignificantly) in the reticulum and omasum, and lowest in the remaining tissues (*p* < 0.01). In calves, the highest expression level was noted in the omasum, while the lowest in the duodenum and jejunum (*p* < 0.01). Furthermore, the *GHSR* expression level in the jejunum and ileum of adult individuals was significantly higher compared to calves (*p* = 0.003, Cohen’s d Effect Size (ES): 1.74, Fold Change (FC): 1.89 for jejunum, and *p* = 0.003, ES: 0.96, FC: 1.90 for ileum) (Figure 1b).

The highest expression level of *SMIM20* in adults was observed in the reticulum, followed by the rumen, and gradually decreased in other segments, with the jejunum exhibiting the lowest transcript level (*p* < 0.01). In calves, the highest expression was noted in the rumen, while the lowest in the jejunum (*p* < 0.01). There were no significant differences in *SMIM20* expression levels between calves and adult individuals (Figure 1c).

In adults, the highest expression level of *GPR173* was found in the omasum, followed by the ileum, while the lowest level was observed in the duodenum (*p* < 0.01). In calves, the highest transcript level was also observed in the omasum, and lowest in the duodenum (*p* < 0.01). In the ileum of adult individuals, *GPR173* expression level was significantly higher compared to calves (*p* = 0.002, ES: 1.77, FC: 2.62). On the contrary, in the colon of calves, *GPR173* expression levels were higher compared to those of adults (*p* < 0.001, ES: 0.60, FC: 2.88) (Figure 1d).

*LEP* expression level in the adult ileum was significantly higher than in other tissues (*p* < 0.01), and this segment, along with the jejunum, also showed significantly higher expression in adult individuals compared to calves (*p* < 0.001, ES: 3.67, FC: 1.41 for ileum, and *p* = 0.008, ES: 1.11, FC: 1.73 for jejunum). The lowest *LEP* expression level in adults was observed in the rumen. On the contrary, the transcript level in the omasum of calves was significantly higher compared to that of adults (*p* < 0.001, ES: 2.29, FC: 2.01) and it was the segment with the highest transcript level in calves (*p* < 0.01) (Figure 1e).

The highest level of *LEPR* was observed in the adult ileum, while the lowest expression was found in the rumen (*p* < 0.01). In calves, the highest mRNA expression was observed in the omasum, while significantly lower mRNA expression was observed in the remaining segments (*p* < 0.01). Significant differences in *LEPR* expression levels were observed between age groups, with higher expression in the omasum of calves compared to adults (*p* < 0.001, ES: 2.27, FC: 3.32), and higher expression in the jejunum of adults (*p* = 0.009, ES: 1.03, FC: 1.92) (Figure 1f).

The expression level of *NUCB2* was relatively consistent among segments, with the lowest expression in the jejunum both in adults and calves (*p* < 0.01). Significant differences in *NUCB2* expression were observed between age groups. In the rumen, the expression of *NUCB2* was higher in calves compared to adult individuals (*p* = 0.008, ES: 2.41, FC: 1.73), while in the ileum *NUCB2* expression was significantly higher in adult individuals (*p* < 0.001, ES: 2.59, FC: 2.05) (Figure 1g).

In summary, as shown in Figure 1h, the interaction between the GIT segment and age, as well as both the GIT segment and age alone, had a significant impact on the relative *GPR173*, *GHSR,* and *LEP* expression levels. In the case of *LEPR* and *NUCB2*, both GIT segment and GIT segment × age interaction had a significant impact on expression levels; finally, age was the only factor significantly affecting *SMIM20* and *GHRL* expression levels.

### 2.2. Correlations between Expression Levels

Multiple significant positive correlations were found among the expression levels of the studied genes in both calves and adults. A thorough analysis is shown in Figure 2a–i. The highest number of positive correlations was observed between the expression levels of *LEP* and *GHSR* in adult individuals. These correlations were present in seven out of the nine examined segments but were not found in the first (rumen) and last (rectum) segments. In calves, the highest number of positive correlations was noted in the omasum, while in adults it was in the abomasum which presented the greatest number of positive correlations. Positive correlations between the ligand (*LEP*) and its receptor (*LEPR*) were identified in the stomach (omasum and abomasum) of both calves and adults, as well as in the reticulum and duodenum of calves. Ligand–receptor correlations were also observed between the remaining pairs: *GHRL/GHSR* in the omasum, jejunum, and rectum of calves, and in the abomasum and jejunum of adults; *SMIM20/GPR173* in the rumen and omasum of calves and the rumen and omasum of adults; *NUCB2/GHSR* in the ileum of calves, and the reticulum and omasum of adults.

### 2.3. Heatmap Analysis

The heatmap analysis showed that, in most GIT segments, gene expression patterns were similar between calves and adults. However, a remarkable disparity in gene expression between the two groups was found in the ileum, as the pattern of genes expression in the adults’ ileum is clustered with genes expression pattern in the omasum of calves and adults. Generally, the highest levels of gene expression were found in the forestomachs and abomasum. No clustering was observed based on the peptide type (orexigenic/anorexigenic) (Figure 3).

## 3. Discussion

In livestock animals, such as domestic cattle, optimal nutrition and consistent weight gain are important factors. These not only impact production efficiency, but further influence the health, longevity, and quality of products such as meat and dairy [15,16]. With such implications, studies on feed intake regulation and related genes are highly valuable [17,18]. While all mammals have a natural mechanism to regulate food intake, it is not uniform across all species. Various factors such as dietary needs, lifestyle, and evolutionary changes explain some of these differences, but some shared mechanisms and communication pathways are present in different mammalian species. In most mammals, the hypothalamus is the main area for the regulation of food intake, with a focus on the arcuate nucleus (ARC) and nucleus tractus solitarus (NTS). These areas process and integrate signals of satiety and hunger from the body, leading to the secretion of various anorexigenic and orexigenic compounds. This then triggers either a catabolic pathway for anorexigenic peptides or an anabolic pathway for orexigenic peptides, which, in turn, involve other brain structures. Hypothalamic neurons in areas responsible for satiety and hunger can be activated by different factors, including regulatory proteins which were the main focus of this study [1,19].

One of the best-known orexigenic proteins is ghrelin, which was isolated from the rat stomach and identified in 1999 by Kojima et al. [20] as a result of their search for a ligand for the orphan receptor GHS-R [20]. It is derived from the precursor protein proghrelin [21]. The earliest known functions of ghrelin include stimulating food intake, influencing lipid metabolism, and regulating growth hormone release [22,23]. Ghrelin also acts locally at GIT to regulate motility and gastric acid secretion [24]. Considering its numerous functions, ghrelin is an important target for potential therapies addressing numerous conditions, including obesity. The initial study on ghrelin expression conducted by its discoverers suggested that the stomach might be its primary source [20]. This pattern of expression has remained consistent for over 20 years, and our results align with it, as the highest expression level was found in the abomasum, followed by the duodenum. However, literature data on the expression of ghrelin in the GIT of domestic cattle are very limited. In a 2020 study, Hayashi et al. [13] examined the level of ghrelin expression in the organs of the GIT of adult Holstein cows and 2-week-old male Holstein calves, simultaneously comparing expression levels between the two age groups. They detected the presence of ghrelin mRNA in all examined sections, namely the rumen, reticulum, omasum, abomasum, duodenum, jejunum, ileum, and colon. However, in both cows and calves, the level of ghrelin expression in the abomasum exceeded the expression levels found in the other organs [13]. Ding et al. [25] conducted an analysis of ghrelin expression in yaks, comparing it with the expression in domestic cattle. However, among the organs analyzed, only two, the abomasum and the duodenum, came from the GIT. The level of ghrelin in the abomasum exceeded the expression level in the duodenum for both cattle and yaks [25]. The level of ghrelin expression in the GIT was also analyzed in another large ruminant, the reindeer. Here, again, expression was observed throughout the entire GIT from the esophagus to the colon, with the highest level in the abomasum, followed by the esophagus and duodenum [26]. Moreover, studies involving sheep, an example of small ruminant, showed the presence of ghrelin mRNA in all chambers of the stomach and in the small intestine, with the highest expression in the abomasum [27,28]. These studies are consistent with similar research conducted on other mammalian species, such as humans [29], pigs [30], mice [31], and guinea pigs [32]. Ding et al. [25] indicated that ghrelin is a protein with significant conservation across species, and, when combined with consistent findings from diverse animal research, this might underscore its pivotal role. The predominant expression of ghrelin in the abomasum implies its potential involvement in managing food consumption in domestic cattle. Furthermore, in our study, *GHRL* expression level did not differ between groups, suggesting that the ghrelin production is not influenced by the developmental stage of the GIT. However, its activity could be modulated by the transcription of its receptor, as we observed that *GHSR* transcription in the small intestine was dependent on age.

Building upon what has been mentioned previously, ghrelin operates via the GHS-R receptor. It is noteworthy that ghrelin activity hinges on the receptor specific variant. The GHS-R splits into two subtypes, GHS-R1a and GHS-R1b, with ghrelin connecting through GHS-R1a [7]. In the available literature, there is a noticeable gap concerning the expression of *GHSR* in the GIT of cattle. However, with respect to human tissues, Ueberberg and his team [29] studied healthy human organs, including sections of the GIT, and found *GHSR* mRNA in the stomach and, to a lesser extent, in the ileum. This is in line with our research, but, in contrast to our study, no expression was found in the colon [29]. On the other hand, studies conducted on young Tibetan and Yorkshire pigs have revealed *GHSR* mRNA expression in all examined organs of the GIT, namely the stomach, small intestine, cecum, and colon. The highest expression levels have been found in the duodenum and jejunum [33]. The expression of *GHSR* has also been studied in rat and guinea pig GIT organs. Expression has been demonstrated in rats, i.e., in the stomach, all sections of the small intestine, colon, and cecum, with the highest level of expression found in the stomach. Interestingly, *GHSR* expression was not found in the same tissues in guinea pigs [34]. Results of our study may indicate that ghrelin, or other GHSR ligands, have a direct effect on the bovine GIT. Furthermore, *GHSR* expression level was higher in the jejunum and ileum of adults compared to those of calves, indicating the potential role of this receptor in nutrient absorption. While the calves we studied have a well-developed digestive tract, the latter is still undergoing growth and development and may differ physiologically from that of adult cattle. Our study suggests that the ghrelin receptor may be one of the factors that develop with age to regulate food absorption.

Anorexigenic nesfatin-1, in spite of its opposing action to ghrelin, shares many features with this peptide. Firstly, it is speculated that nesfatin-1 may act as a ligand for the ghrelin receptor GHS-R1a [35,36,37]. Secondly, both hormones are known to be colocalized in X/A-like cells in the oxyntic glands in the fundus of the stomach, as well as in the pancreas, hypothalamus, and intestine. Additionally, both peptides are formed by enzyme prohormone convertase (PC) 1/3, which converts pro-protein (in the case of nesfatin-1, NUCB2) to their final forms [37]. These similarities, along with their opposing effects on appetite, suggest the need for further research into the interaction between ghrelin and nesfatin-1. As noted in a previous review article, nesfatin-1 has been found to play a protective role in GIT disorders [38]. While its expression in the GIT of cattle has not yet been studied, research has found nesfatin-1 expression in the stomach and intestines of other species, such as rats, mice, and dogs, where the expression level of *NUCB2* in the stomach consistently exceeds that found in the intestines [38,39]. This study is the first to examine *NUCB2* expression across all segments of the digestive tract in domestic cattle, from the rumen to the rectum. The expression level remained relatively consistent and showed a decrease with age only in the rumen. It is worth noting that in calves the development of the forestomachs is concomitant to the initiation of solid food consumption [40]. Consequently, nesfatin-1 may play a supportive role in organ development. In the ileum, higher expression of nesfatin-1 was noted in adult individuals, a phenomenon which could indicate that nesfatin-1 has other or more complex roles in the mature intestine. One aspect that needs further investigation is whether nesfatin-1 affects the expression level of *GHSR*, as it is a possible ligand for *GHSR*. Our study suggests a potential relationship between the two, as both *NUCB2* and *GHSR* expression levels were found to be higher in adults than in calves, possibly due to increased food absorption in adults. Additionally, given that nesfatin-1 can permeate the blood–brain barrier [41], it is plausible to infer that some of this expression facilitates its anorexigenic central action. However, as previously mentioned, there are indications pointing toward nesfatin local activity in the GIT. Therefore, the findings of this study suggest that nesfatin-1 may also exert its local influence on the GIT of domestic cattle and one of the mechanisms may be increased nutrient absorption.

Leptin is a key regulator of the body energy homeostasis, playing an important role in modulating hunger and energy consumption. Primarily produced by adipocytes or fat cells, leptin serves as an indicator to the brain regarding the body fat storage levels. As these levels increase, more leptin is generated, resulting in reduced hunger and increased energy use. On the other hand, when the body fat levels decrease, the production of leptin decreases, causing an increase in appetite and decrease in energy consumption [42]. Interestingly, available studies contrast with ours, as Chelikani et al. [43] noted the absence of *LEP* mRNA in the GIT of calves of domestic cattle [43]. However, it should be noted that in this research PCR products were analyzed semi-quantitatively on agarose gel after electrophoresis, while in the present study real-time PCR was used. Another research endeavor by Hayashi et al. [13] showed the expression of *LEP* mRNA in all sections of the GIT, but only in unweaned calves. In contrast, in adults, the expression was minimal and significantly lower compared to calves [13]. The difference between the present study and that of Hayashi et al. [13] was age and sex of the study subjects: 2-year-old males vs. 5-year-old females (adults) and 7-month-old males vs. 2-week-old males (calves). Furthermore, in a 2002 study, *LEP* expression in the rumen, abomasum, and duodenum was observed only in 3-week-old calves. In older calves and adults, *LEP* mRNA was found only in the duodenum. The same study also showed that, in older calves fed with replacement milk, *LEP* expression also appeared in the rumen and abomasum [44]. Additionally, another study showed a lack of *LEP* expression in the duodenum of dairy cows [45]. This suggests that there may be age, sex, or nutritional-status-related differences that affect the regulation of leptin production in the GIT of cattle. Further research in this area would help better understand these differences and their potential implications for cattle health and production, especially since leptin was present throughout the GIT in our study, including in adult individuals. However, the lack of *LEP* expression in the previous studies is intriguing, as leptin immunoexpression has been found in bovine GIT [13] and *LEP* mRNA has been confirmed to be present in the stomach of rats [46] and humans [47]. In the present study, *LEP* expression level in the omasum of calves was higher than that found in the omasum of adult cattle. As we mentioned earlier, the calves in our study had a relatively well-developed, yet still developing GIT. The forestomachs are the sections where a great deal of change takes place as the ruminant develops. Therefore, further research should investigate the potential function of leptin in the developing calf stomach, as it may play a different role than an anorexigenic peptide. Additionally, in the jejunum and ileum of adult cattle, *LEP* expression level was higher than in calves. Again, as mentioned above, these segments are heavily involved in nutrient absorption; therefore, leptin, along with the ghrelin receptor, may play a role in this process.

The leptin receptor, known as Lep-R or Ob-R, is part of the type I cytokine receptor group and has six different isoforms, specifically Lep-Ra-f [48]. Research in domestic cattle GIT has shown occurrence of the gene expression in sections such as the abomasum, duodenum, jejunum, and ileum [43]. Interestingly, Alam et al. [45] found no expression of the leptin receptor gene in the duodenum of dairy cows, similarly to findings regarding the leptin gene itself [45]. Studies on other species have revealed the presence of *LEPR* mRNA in the human stomach [47] and the stomach and intestines of mice [49]. The present research indicates that the expression pattern of the leptin gene closely matches that of its receptor, suggesting that the ligand might regulate the receptor expression and that leptin may have specific functions in the digestive tract. Studies have demonstrated that leptin may modulate the absorption of macronutrients and influence motility in other species [8]. This is in line with the fact that, in adults, levels of expression for both *LEP* and its receptor *LEPR* were elevated in sections responsible for nutrient absorption. Additionally, both *LEP* and *LEPR* expression levels were significantly higher in the omasum of calves, a phenomenon which may indicate a role played by leptin in the development of this forestomach.

Phoenixin-14, discovered in 2013, emerges from the precursor protein SMIM20 as one of its isoforms. It is known to function as an orexigenic peptide, but its role in the GIT is not well understood. While phoenixin-14 expression at the protein level in the GIT has been demonstrated, there is a lack of literature on its precursor *SMIM20* mRNA expression [50]. Phoenixin-14, which shows considerable homology between species, acts as a ligand for GPR173, a receptor from the G-protein coupled receptor family. This is particularly intriguing, as these receptors are often responsible for relaying important signals [51]. The mRNA expression of *GPR173* in the GIT has not been previously quantified. This study aims to provide the first description of *SMIM20* and *GPR173* expression in the GIT of bovines. The results suggest that *SMIM20* mRNA expression is highest in the initial segments of the digestive tract and gradually decreases. In contrast, *GPR173* expression is highest in the omasum. This could suggest that phoenixin-14, produced mainly in the rumen and reticulum, has a primary effect in the omasum, potentially playing a role in regulating muscular contractions in this segment. However, it is important to note that mRNA levels do not always match protein levels. Therefore, a further investigation is needed to fully understand the role of phoenixin-14 in the bovine digestive system. *GPR173* expression level was higher in adults’ ileum and colon compared to those of calves. This suggests that phoenixin-14 or another unknown ligand for GPR173 may modulate nutrient and/or water absorption.

An interesting observation from the study is that the levels of genes encoding the only two orexigenic peptides examined in the study, ghrelin and phoenixin-14, were not age-dependent (Figure 1h). This could suggest that orexigenic mechanisms in GIT develop earlier in an animal’s life than anorexigenic ones. However, further research is needed to examine this matter, as no such conclusions can be drawn from the results of the two peptides. On the contrary, the expression levels of their receptors were age-dependent (Figure 1h).

In the correlation analysis, a large number of positive correlations in gene expression was noted, which may suggest close working and mutual regulation. Especially in the case of leptin and its receptor, recurring positive correlations were noted. It is worth mentioning that one of the Ob-R isoforms (Ob-Re) can regulate serum leptin concentrations [48], and further studies are needed to examine such relationships in the GIT. Other ligand–receptor correlations were found, as described in the Results sections; however, they differed between groups, while the correlations between leptin and its receptor were similar in both age groups. The most frequent positive correlations were found to occur between the *GHSR* and *LEP* expression levels. This may seem a coincidence, as the *GHSR* encodes a receptor for ghrelin, which in turn acts in opposition to leptin encoded by the *LEP*. However, a study previously cited by us showed a similar pattern of *GHSR* and *LEP* gene expression in pigs [33]. As similar results were obtained in studies with two different species, this aspect needs to be carefully examined in future studies.

The heatmap analysis suggests a potential outlier in the data from the ileum of adult cattle, mainly due to the upregulated expression of *NUCB2*, *LEPR*, and *LEP*. Further investigation is necessary, as mentioned previously in relation to *LEP*. This potential outlier may be related to the role of these proteins in water and nutrients absorption. A speculative hypothesis could be a connection between these peptides and the absorption of vitamin B12 and other fat-soluble vitamins. Given that the ileum plays a key role in this process [52], exploring this possibility could be worthwhile. Additionally, in the heatmap analysis, a clustering between *GHRL* and *NUCB2* can be noted. This is an interesting observation considering the previously mentioned aspects, such as the colocalization of the two peptides encoded by these genes, the sharing of the enzyme, or the potential sharing of the receptor.

Understanding the regulation of food intake in livestock animals, such as cattle, is important for the optimization of meat and dairy production, promotion of sustainable farming, and improvement of animal welfare [53]. Therefore, this aspect is not only significant economically, but also ethically. Furthermore, dairy cows are proposed as a model for the study of food intake in humans, as they share more similarities with humans than the commonly used rodents, e.g., rats or mice. This is because humans and cattle both have a common circadian rhythm, which is significant in nutrition [54].

The limitation of our study is the sample size, which may be considered small. Despite setting a stricter-than-usual significance threshold, this might have influenced our statistical analysis. Nevertheless, a sample size of six or less per group has been previously used in the evaluation of mRNA expression of genes coding for GIT hormones in cattle [13,25], other ruminants [26,27,28,43,44], farm animals [30,33], and even laboratory animals [31,34]. Nonetheless, we believe that this limitation does not completely undermine the novelty of our study. This study, in fact, is the first to report mRNA expression of genes encoding phoenixin-14, nesfatin-1, and GPR173 in the GIT of cattle. We also validated previous reports about the absence of expression of some genes in specific GIT segments in cattle, such as *GHSR* in the colon [29] and *LEP* in the duodenum [45], which, however, were detected in our study. We are of the view that conducting experiments on tissues obtained from healthy cattle during standard culling, specifically from those without pre-existing gastrointestinal conditions and whose carcasses are designated for commercial purposes and consumption, holds significant value for veterinary science, sustainable farming, and human nutrition. Nevertheless, further studies with a larger sample size are recommended to more comprehensively clarify the expression patterns of these genes in domestic cattle of different ages, sexes, or breeds. The accumulation of such data over time can provide deeper insights into the roles of these orexigenic and anorexigenic peptides and their receptors in the GIT of these significant polygastric herbivores.

## 4. Materials and Methods

### 4.1. Animals

The study was conducted on healthy male domestic cattle from the Polish Holstein–Friesian breed aged 20–24 months and weighing 768 ± 46 kg (*n* = 6, adults) and aged 7–8 months weighing 218 ± 23 kg (*n* = 6, calves). All animals, both calves and adults, came from the same farm, lived in the same environment, followed the same feeding regimen, and were the only animals slaughtered that day. The cattle were fed in a semi-intensive system. This entailed a period of grazing on pasture, followed by the total mixed ration (TMR) feeding method [55]. Tissue samples for the study were collected at the local cattle slaughterhouse. The selected animals showed good health, and post-mortem analyses did not reveal any pathologies in the digestive tract. Therefore, the collected material was representative of healthy individuals. The animals were fasted for 18 h before slaughter. Sections of the GIT, namely the rumen, reticulum, omasum, abomasum, duodenum, jejunum, ileum, colon, and rectum, were taken, ensuring that all layers of the respective section were present in the excised fragment (total of 54 samples in each age group). The tissues were rinsed immediately after collection with a physiological saline solution and placed in liquid nitrogen, then frozen at −80 °C. All samples were taken within 15 min of slaughter. According to Polish law, since all tissue collection procedures were conducted post-mortem, ethical review and approval from the Ethics Committee for this study were not required.

### 4.2. Real-Time qPCR Gene Expression Measurement

Tissue fragments were excised from the collected samples, ensuring that they encompassed all layers of the respective section. Subsequently, total RNA was isolated using the PureLink RNA Mini Kit (Invitrogen, Waltham, MA, USA) and following the manufacturer’s provided protocol. The isolated material was further purified from genomic DNA by incubating it with DNase I (PureLink DNase Set; Invitrogen, Waltham, MA, USA). Next, the concentration and purity of the isolated RNA were measured using the NanoDrop 2000 spectrophotometer (Thermo Fisher Scientific, Wilmington, DE, USA), and its integrity was assessed via electrophoresis on a 2% agarose gel. Total RNA in the amount of 300 ng was reverse transcribed into cDNA using the High-Capacity RNA-to-cDNA Kit (Applied Biosystems, Waltham, MA, USA). The resulting cDNA was subjected to real-time PCR analysis to measure the expression of mRNA genes *GHRL*, *GHSR*, *SMIM20*, *GPR173*, *LEP*, *LEPR*, and *NUCB2*, which encode ghrelin, GHS-R, phoenixin-14, GPR173, leptin, LEP-R, and nesfatin-1, respectively. *ACTB* and *RPS9* were used as housekeeping genes [56]. The primer sequences (see Table 1) were designed in Primer3web [57] and obtained from Genomed (Genomed, Warszawa, Poland). The primers for each qPCR amplicon were designed to span introns and were ideally located in different exons to prevent DNA amplification. Real-time PCR analysis was performed on a QuantStudio 7 Flex (Applied Biosystems, Waltham, MA, USA), utilizing RT PCR Mix SYBR Green with HiROX, according to the protocol (A&A Biotechnology, Gdańsk, Poland). Each sample was subjected to three technical replicates for the reactions. The parameters were as follows: pre-run at 95 °C for 3 min, 45 cycles with a denaturation step at 95 °C for 15 s, annealing at 57 °C for 30 s and an extension step at 72 °C for 30 s. The amplicon efficiency was determined by analyzing the slope coefficient of the standard curve derived from serial dilutions of pooled cDNA (E = efficiency (10^[−1/slope]^). Using the −ΔΔCT method, the relative expression of *GHRL*, *GHSR*, *SMIM20*, *GPR173*, *LEP*, *LEPR*, and *NUCB2* genes was calculated [58].

### 4.3. Statistical Analysis

Statistical analyses were conducted using the Statistica software (v. 13.0, TIBCO Software Inc., Palo Alto, CA, USA), GraphPad Prism software (v. 10.1.0, GraphPad Software, San Diego, CA, USA), and R software, v. 4.3.1 [59]. To evaluate differences in mRNA expression across GIT sections, relative quantification data were log2 transformed for normalization, and the data normality was confirmed using the Shapiro–Wilk test. The data were analyzed using a two-way ANOVA, with the general linear model including each gene as a dependent variable and GIT segment, age, and their interaction as the independent effects. Model residuals were tested to validate the assumption of normality and homoscedasticity using QQ-plots, the Shapiro–Wilk test, and residual–fitted values plots. Tukey’s HSD post hoc test, as recommended by the GraphPad software, was applied for correction of multiple comparison tests using statistical hypothesis testing. A multiplicity-adjusted *p*-value < 0.01 was considered statistically significant, accounting for multiple comparisons against a family-wise alpha error threshold set to 0.01 (99% confidence interval). This stricter-than-usual threshold was chosen to limit the number of statistically significant findings. Additionally, Cohen’s d was used as an indicator of effect size of all significant differences in gene expression within specific GIT segments between calves and adults.

Correlation plots illustrating the expression patterns of the analyzed genes in GIT sections were generated using the corrplot package in R. Significant correlations are indicated by asterisks (* *p* < 0.05, ** *p* < 0.01, *** *p* < 0.001). The heatmaps depicting gene expression data were produced using the pheatmap package in R, after scaling the fold change profiles of each gene individually using the z-score formula (z-score = (individual gene value − mean gene value)/SD).

## 5. Conclusions

This study revealed consistent mRNA presence for all examined genes across various GIT segments in both calves and adult cattle, emphasizing their importance in GIT organs. Notable expression differences were observed between the two age groups, particularly in the rumen, omasum, jejunum, ileum, and colon, suggesting that peptide activity regulating food intake varies with developmental stages. In adults, heightened expression of *GHSR*, *GPR173*, *LEP*, and *NUCB2* in the small and large intestines implies they play a role in nutrient absorption. Conversely, elevated levels of *LEP*, *LEPR*, and *NUCB2* in the omasum and rumen of calves underline their possible importance in the development of bovine forestomachs. The study also found that the expression of orexigenic peptides *GHRL* and *SMIM20* is not dependent on age, suggesting more uniform mechanisms for orexigenic processes compared to anorexigenic ones.

## Figures and Tables

**Figure 1 ijms-25-00533-f001:**
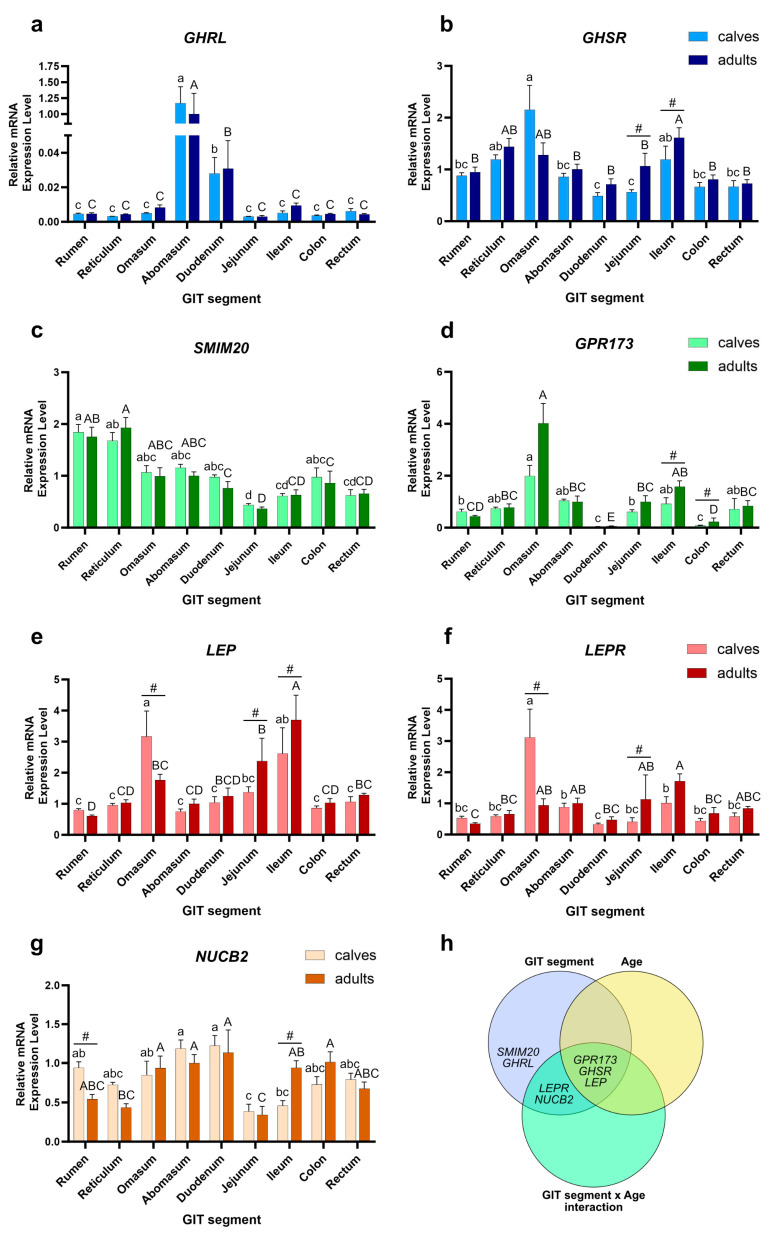
Gene expression levels. The relative expression of genes (**a**) *GHRL*, (**b**) *GHSR*, (**c**) *SMIM20*, (**d**) *GPR173*, (**e**) *LEP*, (**f**) *LEPR*, and (**g**) *NUCB2* in the examined bovine GIT segments, (**h**) Venn diagram summarizing the genes measured for which GIT segment, age, or their interaction had a significant impact on their relative mRNA expression levels as tested by the two-way ANOVA. Expression was normalized using the geometric mean of housekeeping genes *RPS9* and *ACTB* and is presented relative to the levels observed in the adult abomasum. Different lowercase letters denote significant differences between GIT segments in calves, while different uppercase letters denote significant differences between GIT segments in adult individuals (*p* < 0.01). A hash (#) highlights significant differences in mRNA expression between calves and adults within a specific GIT segment (*p* < 0.01). Owing to the exponential characteristics of relative mRNA expression computed via the −ΔΔCT method, the geometric means along with SE (standard errors) are presented.

**Figure 2 ijms-25-00533-f002:**
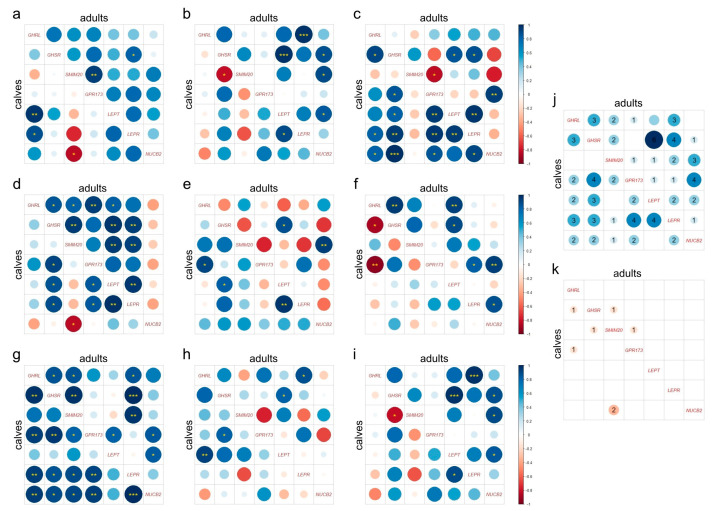
Correlation plot analysis. The correlation among expression patterns for *GHRL*, *GHSR*, *SMIM20*, *GPR173*, *LEP*, *LEPR*, and *NUCB2* in (**a**) rumen, (**b**) reticulum, (**c**) omasum, (**d**) abomasum, (**e**) duodenum, (**f**) jejunum, (**g**) ileum, (**h**) colon, and (**i**) rectum, for both calves and adult individuals. The correlation type is indicated by the color, with blue representing a positive correlation and red representing a negative correlation. The size of the dots reflects the magnitude of the correlation coefficient. Statistically significant correlations are marked with asterisks (* *p* < 0.05, ** *p* < 0.01, *** *p* < 0.001). The total number of significant (**j**) positive and (**k**) negative correlations among the expression levels of the seven analyzed genes in the GIT segments for calves and adults is presented.

**Figure 3 ijms-25-00533-f003:**
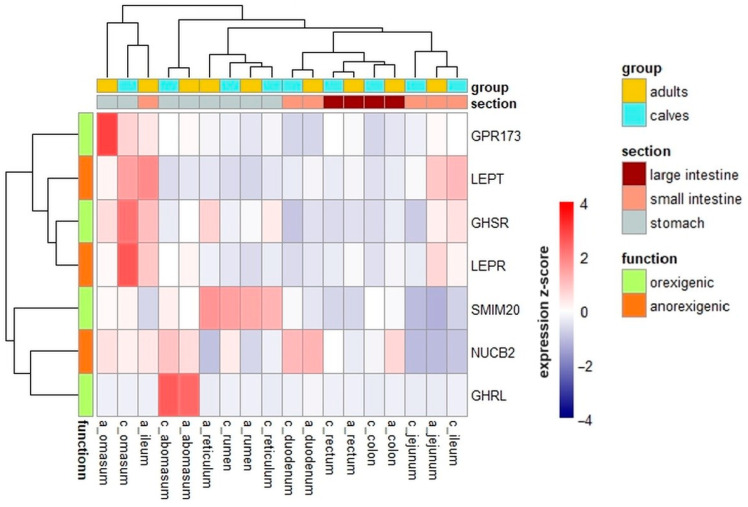
Heatmap and dendrogram of hierarchical clustering based on the z-score of gene expression. The heatmap illustrates the GIT segment-specific expression patterns of *GHRL*, *GHSR*, *SMIM20*, *GPR173*, *LEP*, *LEPR*, and *NUCB2* genes in both calves and adult individuals. Expression levels are represented as z-scores calculated per row (per gene) to highlight variations in the average expression of each gene across the GIT segments in both calves and adults. The *x*-axis represents sampled tissues, using the prefix “c” for calves and “a” for adults.

**Table 1 ijms-25-00533-t001:** Primers used in the study.

Gene	Primer Sequences (5′ to 3′) ^1^	Product Length (bp)	GeneBank Accession Number	qPCR Efficiency
*GHRL*	F: ^133^ TCAGGCAGACTGAAGCCCCG R: ^223^ GGATTTCCAGCTCGTCCTCTGC	91	NM_174067.2	1.98
*GHSR*	F: ^622^ CGCTCCGGACTGCTCACAGT R: ^842^ AAGGGCAGCCAGCAGAGGAT	221	NM_001143736.2	2.00
*SMIM20*	F: ^229^ GCCATAAATCGAGCTGGTAT R: ^376^ TGCTGCAGAACTGAAAGCAT	148	NM_001145428.1	1.96
*GPR173*	F: ^843^ GCAAGATTGTGGCCTTTATGGCTG R: ^961^ CATGCGCTTGGCATAGAAG	119	NM_001015604.1	1.83
*LEP*	F: ^46^ AAATGCGCTGTGGACCCCTGT R: ^245^ GAGCCCAGGGATGAAGTCCAA	200	NM_173928.2	2.00
*LEPR*	F: ^1781^ AATCTGCCAGTCTCCCAGTG R: ^1897^ CAACTGTGTGGGCTGGAGTA	117	NM_001012285.2	1.97
*NUCB2*	F: ^276^ AAAAGCTCCAGAAAGCAGACA R: ^393^ GCCACTTCTTGCCTTTTCAG	118	NM_001075381.1	1.98
*ACTB*	F: ^795^ TCCCTGGAGAAGAGCTACGA R: ^927^ AGGTAGTTTCGTGAATGCCG	133	NM_173979.3	2.02
*RPS9*	F: ^128^ CCTCGACCAAGAGCTGAAG R: ^191^ CCTCCAGACCTCACGTTTGTTC	64	NM_001101152.2	2.00

^1^ The numbers indicate the position at which the respective primers bind to the target genes (bp in 5′ to 3′ direction).

## Data Availability

Data will be made available on the request.

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
