# Peer review of "Expression of Genes Encoding Selected Orexigenic and Anorexigenic Peptides and Their Receptors in the Organs of the Gastrointestinal Tract of Calves and Adult Domestic Cattle (Bos taurus taurus)"

_ijms, 2023, doi:10.3390/ijms25010533_

Round 1

Reviewer 1 Report

Comments and Suggestions for Authors

The research subject meets the aims and scopes of the journal, but I am concerned about the quality of the dataset, which in my opinion questions the merit of the study. Please, find below a discussion of the most critical points.

·         Study goal. Regulation of appetite and feed intake is a rather complex and the selection of target genes appears limited. In addition, transcriptomics of said genes is only one side of the medal and it must not be that protein expression or activity of said proteins follows respective trends obtained with targeted transcriptomics. ELISA assays accompanying the gene expression results are be the minimum that must be presented in this case, in order to allow some conclusion on the biological relevance of the findings.

·         Spectrum of measurements. The phenotyping of animals appears rather crude. In fact, only the transcriptomics are presented. Given the nature of the study subject, slaughter weight, fatness of carcasses, and analysis of the contents of respective GIT sections for nutrients should have been provided to allow a better comparison between animals and identify potential biasing factors (e.g. higher fiber intake may affect these genes as does a higher energy density etc.).

·         Animals. Sample size is rather limited especially since animals were not part of an experimental design but randomly chosen at the slaughterhouse. This means animals probably originated from different farms, hence different environments and feeding conditions, which most likely affect the gene expression patterns under study by differences in epigenetic imprinting as well as diet-host-microbe interaction. I am concerned that any conclusions on the general importance of the expression of these genes in certain parts of the digestive tract and between age groups may therefore be biased by factors which cannot be controlled for. Since animals are too low in number to establish respective statistical classes and also too low in number to be representative for the population of Polish HF cattle or cattle in general, obtained data and conclusions arising thereof are of limited reliability.

·         qPCR methodology. Authors chose reference genes based on an earlier study. This is against good practice, which dictates that in each new experiment reference genes must be again be “cherry-picked” and quality-checked as one cannot assume that they are always not regulated even when repeating the same experiment. Special caution is advisable in the present case, where random animals were sampled in a slaughterhouse, without knowledge of their physiological status. Finally, the normalization method by Livak and Schmittgen assumes 100% amplification efficiency. This is not always the case, which is why amplification efficiency should at least be checked for applied assays and, if applicable, acknowledged for normalization purposes by applying a suitable model. Gold-standards for conducting and communicating qPCR experiments (including selection of reference genes) can be found in the following papers for example: https://doi.org/10.1373/clinchem.2008.112797 , https://doi.org/10.1007/978-1-4939-0733-5_3 , https://doi.org/10.1093/clinchem/hvab052

·         Table 1. It would be good to also state the region (as bp in 5'-->3' direction) where respective primers are supposed to bind the target or reference genes, respectively.

·         Statistics. Analyzing GIT segment effects independent of the age effects appears questionable and it should be defended why it was not considered to build a general linear model or mixed model analyzing segment, age, and interaction in a combined dataset.

It could also be argued if P<0.05 is truly the right threshold considering the rather low sample size which probably does not yield enough power (1-beta = 0.8 is usually considered the minimum) at alpha = 0.05.

Furthermore, even if 0.05 was the correct threshold or if authors would correct the threshold down to be more strict on the definition of "significance", the reliability of such a result is limited due to the low sample size. It is well accepted that studies with low statistical power have a lower chance of detecting a true effect, but it is less well acknowledged that low power also reduces the likelihood that a statistically significant result reflects a true effect. The probability that a research finding reflects a true effect decreases as statistical power decreases for any given pre-study odds and a fixed type I error level. Though, more effect size and, hence, statistical power requires higher sample sizes of animals, an issue some may refer to as unethical from an animal rights point of view, but the number of animals wasted year by year by underpowered non-reproducible studies is much higher. Therefore, the true ethics issue arises from underpowered studies. These issues are addressed in much more detail in the following paper, which I urgently recommend everyone to study in detail. Though the authors deal with the case of power in neuroscience, the general implications from this manuscript are relevant for all types of animal studies. https://doi.org/10.1038/nrn3475

Comments on the Quality of English Language

Errors in English writing are evident throughout the manuscript, although the writing style in general is quite well.

Author Response

Reviewer #1

We would like to express our sincere gratitude to the Reviewer for their time, support, and constructive comments. We have carefully read their suggestions and have made the necessary corrections that we hope will meet the Reviewer's approval. We have highlighted the revisions in the manuscript. In response to the Reviewer's comments, we have addressed each point as follows:

  1. Study goal. Regulation of appetite and feed intake is a rather complex and the selection of target genes appears limited. In addition, transcriptomics of said genes is only one side of the medal and it must not be that protein expression or activity of said proteins follows respective trends obtained with targeted transcriptomics. ELISA assays accompanying the gene expression results are be the minimum that must be presented in this case, in order to allow some conclusion on the biological relevance of the findings.

Thank you for this comment. Our main objective was to confirm the expression of selected genes in the gastrointestinal tract, which, in part, had not been done previously in cattle (the expression of NUCB2, SMIM20, GPR173 was determined for the first time), or had been performed only for selected sections of the gastrointestinal tract (LEP, LEPR, GHRL, GHSR), or allowed us to validate previous reports of the absence of expression of some genes in specific tissues (GHSR in the colon, LEP in the duodenum, and others), whose expression in these GIT sections, however, we did detect.

We agree that studying the transcriptome alone is not sufficient to fully describe the biological significance of the results. It should be noted, as the reviewer pointed out, that the transcriptome does not directly translate into protein, and there are numerous and complex mechanisms regulating the post-transcriptional fate of mRNA. Furthermore, in the case of some peptides, such post-translational changes can lead to the production of mutually exclusive peptides (like PNX-14 and PNX-20 for SMIM20). Moreover, ELISA would measure the total protein level in the tissue, including that which was not transcribed from mRNA in that tissue.

We would also like to point out that the presented work is part of a broader study. We are currently preparing additional publications, in which, based on histological IHC determinations, we will describe in detail the localization and colocalization of individual peptides and their receptors in the structures of various GIT segments. Since IHC determinations strictly involve the study of proteins, we decided that for these studies, to support the qualitative determinations of protein immunoreactivity, we will use quantitative ELISA assays or semi-quantitative WB.

It is worth mentioning that this approach has previously been considered appropriate by other researchers, who in their publications chose to present an exclusive analysis of the expression of genes without determining protein levels: https://doi.org/10.3390/ijms22084038, https://doi.org/10.3892/mmr.2015.3939, https://doi.org/10.1590/1806-9061-2016-0344, https://doi.org/10.1016/j.ygcen.2015.05.004, https://doi.org/10.1016/S0739-7240(00)00064-3, https://doi.org/10.1017/S1751731117001410, https://doi.org/10.1016/S0165-2427(02)00260-X,  https://doi.org/10.1007/s11033-023-08501-6, https://doi.org/10.3168/jds.2007-0763, https://doi.org/10.3168/jds.2011-5311, https://doi.org/10.1016/S0960-0760(03)00025-6, https://doi.org/10.1186/1471-2164-12-140,  

  1. Spectrum of measurements. The phenotyping of animals appears rather crude. In fact, only the transcriptomics are presented. Given the nature of the study subject, slaughter weight, fatness of carcasses, and analysis of the contents of respective GIT sections for nutrients should have been provided to allow a better comparison between animals and identify potential biasing factors (e.g. higher fiber intake may affect these genes as does a higher energy density etc.).

Thank you for your comment. We understand the importance of the factors you've mentioned, particularly in relation to our study's subject. Our primary focus was to ensure group homogeneity in our experiment. To this end, all animals used in our study came from a single breeder, ensuring consistent conditions in terms of animal husbandry, nutrition, age, and fasting time before slaughter. We have provided more details about their origin, nutrition, and weight in the revised manuscript (lines 427-435).

Regarding the suggestion to analyze digestive tract content and carcass fatness, these are indeed important factors. While fat content can influence the secretion of leptin and other appetite-regulating peptides, many variables could potentially affect our results. Our study primarily aimed to evaluate peptide expression levels with respect to two factors: GIT section and age, using two homogeneously grouped sets of cattle. The cattle were fed using a semi-intensive system, which is generally preferred over intensive systems for reasons of animal welfare. This feeding approach typically results in moderate carcass fatness. However, the fat levels were similar across all animals in each group, suggesting it didn't significantly impact our findings.

When planning our experiment, we conducted an extensive review of the current literature and adhered to established guidelines. Our review of scientific literature on ruminants indicated that such detailed analyses of digestive tract contents and carcass fatness are not standard practice in studies of this nature. This conclusion is supported by studies such as https://doi.org/10.1093/tas/txaa019, https://doi.org/10.3168/jds.2010-3205, and https://doi.org/10.3168/jds.2011-5311.

  1. Sample size is rather limited especially since animals were not part of an experimental design but randomly chosen at the slaughterhouse. This means animals probably originated from different farms, hence different environments and feeding conditions, which most likely affect the gene expression patterns under study by differences in epigenetic imprinting as well as diet-host-microbe interaction. I am concerned that any conclusions on the general importance of the expression of these genes in certain parts of the digestive tract and between age groups may therefore be biased by factors which cannot be controlled for. Since animals are too low in number to establish respective statistical classes and also too low in number to be representative for the population of Polish HF cattle or cattle in general, obtained data and conclusions arising thereof are of limited reliability.

Thank you for this comment. When planning our studies, we ensure that experimental samples, even those not obtained under the direct oversight of an ethical committee, comply with European Union law regarding the protection of animals used for experimental or scientific purposes. Additionally, these samples are experimentally valuable in terms of current scientific knowledge and recommendations. Therefore, when designing this study, we reviewed existing research on the topic (gene expression analysis in bovine tissues and histological examination of the gastrointestinal tract in healthy cattle with experimental factors like breed, sex, age) in Scopus and PubMed databases. In the majority of these studies, a group size of 3-6 animals was used (please see the list given belove), due to methodological reasons and practical limitations, as detailed below.

In veterinary research involving live animals on cattle farms, considering cattle's general physiology and  metabolic rate, to ensure homogeneity within the experimental group, all samples should be collected within a three-hour timeframe. Collecting samples, including gastrointestinal tract sections, from large farm animals, even during standard slaughterhouse procedures, entails additional, specific limitations. The presence of certain personnel in the slaughterhouse and the type and transport of collected tissues require approval from veterinary authorities. The pace of slaughter and dissection depends on the facility's efficiency and procedures, not the researchers, and differs significantly from that for pigs, poultry, or laboratory animals. However, the same three-hour timeframe applied on the farm must be strictly adhered to in the slaughterhouse, meaning the entire process of slaughtering, dissecting, and tissue collection from all animals must fit within this period. This considerably limits the number of animals from which tissues can be collected and preserved, explaining why the number of animals in cited studies ranged from 3 to 6 per group.

Bearing these limitations in mind, special attention was paid to using animals with similar phenotypes in our study. While our manuscript did not specify that animals were randomly selected from a slaughterhouse, we did not clarify the selection process, which has been corrected in the revised manuscript (lines 428-430). As indicated before, all animals, both calves and adults, came from the same farm, lived in the same environment, followed the same feeding regimen, and were the only animals slaughtered at the facility that day. The entire slaughter process fell within the designated three-hour window, and all samples were taken, cleaned, and snap-frozen in liquid nitrogen within 15 minutes after the animal's death.

In conclusion, we believe the number of experimental animals used was typical for this type of cattle study and allowed for proper statistical analysis and comparison with similar previous studies. We hope this explanation convinces the Reviewer that our study adhered to all fundamental principles of planning and conducting experiments on healthy cattle as farm animals.

Reviewed previous works on gastrointestinal tract of cattle, in which 3-6 animals per group were used: 10.3168/jds.2010-3205, 10.3168/jds.2011-5311, 10.1007/s00441-013-1570-5, 10.1007/s00418-006-0250-x, 10.1007/s11259-006-3216-5, 10.1007/s10142-012-0308-x, 10.3168/jds.2021-21355, 10.2460/ajvr.67.12.1992, 10.1177/030098581348295, 10.1016/j.regpep.2011.12.004, 10.3168/jds.S0022-0302(03)73830-2, 10.1073/pnas.1820130116, /j.1439-0264.2003.00456.x, 10.1007/BF03195635, doi.org/10.3168/jds.2012-5506, 10.1292/jvms.19-068, 10.1002/j.1939-4640.2000.tb03396.x, 10.1016/S0960-0760(03)00025-6, 10.1016/s0739-7240(01)00114-x, 10.1371/journal.pone.0092592, 10.1016/j.domaniend.2011.09.006, 10.1292/jvms.19-0680, 10.1111/ahe.12400.

  1. qPCR methodology. Authors chose reference genes based on an earlier study. This is against good practice, which dictates that in each new experiment reference genes must be again be “cherry-picked” and quality-checked as one cannot assume that they are always not regulated even when repeating the same experiment. Special caution is advisable in the present case, where random animals were sampled in a slaughterhouse, without knowledge of their physiological status. Finally, the normalization method by Livak and Schmittgen assumes 100% amplification efficiency. This is not always the case, which is why amplification efficiency should at least be checked for applied assays and, if applicable, acknowledged for normalization purposes by applying a suitable model. Gold-standards for conducting and communicating qPCR experiments (including selection of reference genes) can be found in the following papers for example: https://doi.org/10.1373/clinchem.2008.112797, https://doi.org/10.1007/978-1-4939-0733-5_3, https://doi.org/10.1093/clinchem/hvab052

All RNA-related laboratory procedures were conducted in accordance with good laboratory practices recommended by Pfaffl, and detailed in Nolan et al. “Good practice guide for the application of quantitative PCR (qPCR)”, LGC 2013 (https://www.gene-quantification.de/national-measurement-system-qpcr-guide.pdf). To ensure RNA quality, we checked for protein or chemical contamination using a spectrophotometer (A260/280 ratio). Only RNA samples that were not degraded (next step - agarose gel quality control) or contaminated were used for further analysis; otherwise, the isolation was repeated. Although DNase treatment is optional, it was performed on each sample to minimize DNA contamination. Additionally, what is very important, the primers for each qPCR amplicon were designed to span introns and were ideally located in different exons to prevent DNA amplification.  This information has been added to the manuscript (lines 459-461). We conducted PCR amplification tests on both DNA and cDNA samples and confirmed amplification only for cDNA.

While the manuscript originally described using the general -ddCT method for fold change calculations and cited the work of Livak and Schmittgen, we actually implemented Pfaffl's correction, which accounts for amplicon efficiency, which was calculated based on the standard curve's slope and included in the RQ value calculations. This clarification has been added to the revised Materials and Methods section, along with the appropriate citation of Pfaffl's work (10.1093/nar/29.9.e45) (lines 468-469) and information about amplicons’ efficiency in Table 1. Our studies consistently employ the Pfaffl method, as evidenced by our publications, including 10.3390/ijms24097715, 10.2478/aoas-2022-0028, 10.3390/cells11081268, and 10.2478/aoas-2022-0001.

Regarding the selection of housekeeping genes (hkg), the expression of both selected hkgs was stable across all tested tissue types what was supported at manuscript by citation (line 655).  Moreover, these hkgs were previously verified as the most stable in healthy bovine tissues, including the gastrointestinal system, and are widely used, as demonstrated in numerous studies (10.3168/jds.2006-640, 10.3390/ani11020563, doi.org/10.1152/physiolgenomics.00102.2011, 10.1093/jas/sky016, 10.1152/physiolgenomics.00223.2006, 10.1371/journal.pone.0191558, 10.1111/ahe.12128, 10.1371/journal.pone.0142633, 10.1093/jas/skab116, 10.1038/s41598-022-07682-7)

  1. Table 1. It would be good to also state the region (as bp in 5'-->3' direction) where respective primers are supposed to bind the target or reference genes, respectively.

Thank you for this valuable suggestion. This information has been added to Table 1.

  1. Analyzing GIT segment effects independent of the age effects appears questionable and it should be defended why it was not considered to build a general linear model or mixed model analyzing segment, age, and interaction in a combined dataset.

We have revised the statistical analysis of gene expression, employing a different approach. The data were analyzed using two-way ANOVA, with the general linear model including gene as a dependent variable and GIT segment, age, and their interaction as independent effects. Model residuals were examined for assumptions of normality and homoscedasticity using QQ-plots, the Shapiro–Wilk test, and residual-fitted values plot. Tukey’s HSD post hoc test was applied for the correction of multiple comparison tests using statistical hypothesis testing. Additionally, we performed analyses using the Bonferroni correction for multiple comparisons, which yielded similar results to Tukey’s HSD. Consequently, we decided to continue using Tukey’s HSD, as recommended by GraphPad software. To emphasize the significance of the main effects and interactions in the employed models, a Venn plot corresponding to the results has been included.

  1. It could also be argued if P<0.05 is truly the right threshold considering the rather low sample size which probably does not yield enough power (1-beta = 0.8 is usually considered the minimum) at alpha = 0.05. Furthermore, even if 0.05 was the correct threshold or if authors would correct the threshold down to be more strict on the definition of "significance", the reliability of such a result is limited due to the low sample size. It is well accepted that studies with low statistical power have a lower chance of detecting a true effect, but it is less well acknowledged that low power also reduces the likelihood that a statistically significant result reflects a true effect. The probability that a research finding reflects a true effect decreases as statistical power decreases for any given pre-study odds and a fixed type I error level. Though, more effect size and, hence, statistical power requires higher sample sizes of animals, an issue some may refer to as unethical from an animal rights point of view, but the number of animals wasted year by year by underpowered non-reproducible studies is much higher. Therefore, the true ethics issue arises from underpowered studies. These issues are addressed in much more detail in the following paper, which I urgently recommend everyone to study in detail. Though the authors deal with the case of power in neuroscience, the general implications from this manuscript are relevant for all types of animal studies. https://doi.org/10.1038/nrn3475

Thank you for your valuable comment. In the revised statistics, instead of the previously selected P < 0.05, an adjusted P-value < 0.01 was considered statistically significant. This adjustment accounts for multiple comparisons against a family-wise alpha error threshold set to 0.01, corresponding to a 99% confidence interval. Additionally, Cohen's d was utilized as an indicator of the effect size for all significant differences in gene expression within specific GIT segments between calves and adults.

In our research, when feasible, we always perform a priori estimation of group sizes based on power analysis (refer to our recent works: 10.3390/foods12203733, 10.3390/ijms24097715, 10.2478/aoas-2022-0022, 10.3390/jcm11092441, 10.3390/ani11123556, 10.3390/ani11051468, 10.3390/ani11051349). Due to reasons explained previously (see our detailed response to point 3), our ability to directly follow a priori sample size calculations and conduct the study on a larger number of animals was limited. Nevertheless, the number of experimental animals in our study meets the minimum requirements as determined by the “resource equation” method, described by Mead in “The design of experiments” (1988) and by Festing and Altman in “Guidelines for the design and statistical analysis of experiments using laboratory animals” (10.1093/ilar.43.4.244). According to this method, the minimal number of animals per group for a 2-group experiment, considered adequate, is 6, a criterion our study fulfills. We have, however, acknowledged the limited sample size as a study limitation at the end of the discussion section.

We wish to reiterate that our primary research aim was not to investigate changes in mRNA expression in nutritional or toxicological studies, pathological tissues, nor to conduct pre-clinical experiments in medical sciences where cattle, as polygastric animals, are not suitable model animals. Our experiment is primarily situated in veterinary science and biology and also falls within the scope of animal production sciences. As mentioned in the introduction, understanding the mRNA expression levels of different anorexigenic and orexigenic peptides in the GIT of polygastric animals, which significantly differs from monogastric animals, provides valuable insights for future studies on the complex mechanisms governing feeding behavior and feed intake in polygastric herbivores. However, before delving deeper, an initial, even simple study, must be conducted. We believe that conducting an experiment on tissues obtained from healthy cattle during standard culling, specifically those without pre-existing gastrointestinal conditions and whose carcasses are designated for commercial use and consumption, would be valuable for veterinary science, sustainable farming, and human nutrition. This holds true even if the experiment is performed with lower statistical power.

As veterinarians, our foremost priorities are always the welfare and health of animals.

Reviewer 2 Report

Comments and Suggestions for Authors

This paper explores the mRNA expression levels of genes related to appetite regulation in the gastrointestinal tract (GIT) of domestic cattle, particularly the presence of ghrelin, nesfatin-1, phoenixin-14, leptin, and their respective receptors.

Suggestions for Improvement:

In the Introduction section, highlight the specific questions this study aims to address and provide a clear hypothesis or research questions.

In the Discussion section, emphasize the biological significance of the gene expression differences observed in the different GIT segments and between calves and adults.

In the Materials and Methods section, the PCR cycling conditions should be added.

Overall, the paper presents a valuable contribution to understanding appetite regulation in domestic cattle. However, addressing the above suggestions would enhance the clarity and impact of the study.

Author Response

We would like to express our sincere gratitude to the Reviewer for their time, support, and constructive comments. We have carefully read their suggestions and have made the necessary corrections that we hope will meet the Reviewer's approval. We have highlighted the revisions in the manuscript. In response to the Reviewer's comments, we have addressed each point as follows:

This paper explores the mRNA expression levels of genes related to appetite regulation in the gastrointestinal tract (GIT) of domestic cattle, particularly the presence of ghrelin, nesfatin-1, phoenixin-14, leptin, and their respective receptors.

Suggestions for Improvement:

In the Introduction section, highlight the specific questions this study aims to address and provide a clear hypothesis or research questions.

Thank you for this comment. A clearer aim of the study has been added to the revised version of the manuscript in lines 77-82.

In the Discussion section, emphasize the biological significance of the gene expression differences observed in the different GIT segments and between calves and adults.

Thank you for this comment. We added more information about the biological significance of the genes expression in the revised version of the manuscript in lines 267-269, 286-298, 329-334, 341-349, 365-368, 391-394.

In the Materials and Methods section, the PCR cycling conditions should be added.

Thank you for this comment, PCR conditions have been added to the manuscript in lines 464-466.

Overall, the paper presents a valuable contribution to understanding appetite regulation in domestic cattle. However, addressing the above suggestions would enhance the clarity and impact of the study.

Thank you for your positive feedback.

Reviewer 3 Report

Comments and Suggestions for Authors

In their manuscript Kras and co-worker report a descriptive and relatively simple work in which they asses the expression levels of different well-chosen genes, putatively involved in the appetite control of mammals. Gene expression is analysed along different sections of the GIT in 6 adults cows (20-24 months) and 6 calves (7-8) (sampled in the slaughterhouse).  Despite their simplicity in the design, the manuscript is extremely well written and undoubtedly provide the readers with an actualized state of the art, identifying main hot topics or lacks of knowledge, and also make a really enlightening discussion of the results.

Despite lost of merits in the manuscript, there are still some issues that would deserve to be improved. Particularly it would be recommended to complement the statistical analysis with additional approaches.  Regarding the comparison of gene expression across sections and between calves and adults, it would be suggested to make an integral analysis with a two-way ANOVA in which you will be able to analyse the possible interaction between sections and ages. Regarding the correlation analysis it could be really interesting to analysis possible correlations between signal proteins and receptors along different sections of the GIT. Did you analyse for example possible correlations between GHRL in abomasum and GHSR in ileum? Or between phoenixin-14 in rumen/reticulum with phoenixin-14 in omasum? May be these kinds of connections could improve the discussion and point out possible mechanisims of regulation.

Specific comment.

L81-L83. Can be also interesting to highlight not only the intensive growth of calves but the relevant transformation from a pre-ruminant to a ruminant condition with high implication in physiology and functions of the GIT.  

L64-L89. A bit long this part. Probably can be summarize a little bit.

L143. “No significant recurring ligand-receptor type correlations were identified”. Did you test suck kind of correlations between different  GIT sections? (comment above).

L157-159. Review writing of these two sentences. As now they seem contradictory.

L159. Differences in reticulum between calves and adults do not seem relevant. They cluster together.

L330-334. Four lines for discussion of correlation analysis and heatmap seems scarce. Particularly it is suggested to improve the correlation analysis approach and probably in this way the discussion (see general comment above).

L338-346. Too long paragraph?

L350. It can be relevant to provide information regarding the feeding regimen of these animals and production system (to evaluate p.e. the development of the rumen fermentative capacity in calves, or the possible impact of more or less forage in the diet) and also the management of the animals previous to slaughter (feed withdrawal time, transport…etc) that could determine relevant changes in the short-time metabolic control of appetite.  These elements would also deserve some discussion.

Author Response

We would like to express our sincere gratitude to the Reviewer for their time, support, and constructive comments. We have carefully read their suggestions and have made the necessary corrections that we hope will meet the Reviewer's approval. We have highlighted the revisions in the manuscript. In response to the Reviewer's comments, we have addressed each point as follows:

In their manuscript Kras and co-worker report a descriptive and relatively simple work in which they asses the expression levels of different well-chosen genes, putatively involved in the appetite control of mammals. Gene expression is analysed along different sections of the GIT in 6 adults cows (20-24 months) and 6 calves (7-8) (sampled in the slaughterhouse).  Despite their simplicity in the design, the manuscript is extremely well written and undoubtedly provide the readers with an actualized state of the art, identifying main hot topics or lacks of knowledge, and also make a really enlightening discussion of the results.

Thank you for your positive feedback.

Despite lost of merits in the manuscript, there are still some issues that would deserve to be improved. Particularly it would be recommended to complement the statistical analysis with additional approaches.  Regarding the comparison of gene expression across sections and between calves and adults, it would be suggested to make an integral analysis with a two-way ANOVA in which you will be able to analyse the possible interaction between sections and ages. Regarding the correlation analysis it could be really interesting to analysis possible correlations between signal proteins and receptors along different sections of the GIT. Did you analyse for example possible correlations between GHRL in abomasum and GHSR in ileum? Or between phoenixin-14 in rumen/reticulum with phoenixin-14 in omasum? May be these kinds of connections could improve the discussion and point out possible mechanisims of regulation.

Thank you for this comment. We have revised the statistical analysis of gene expression, employing a different approach. The data were analyzed using two-way ANOVA, with the general linear model including gene as a dependent variable and GIT segment, age, and their interaction as independent effects. Model residuals were examined for assumptions of normality and homoscedasticity using QQ-plots, the Shapiro–Wilk test, and residual-fitted values plot. Tukey’s HSD post hoc test was applied for the correction of multiple comparison tests using statistical hypothesis testing. Additionally, we performed analyses using the Bonferroni correction for multiple comparisons, which yielded similar results to Tukey’s HSD. Consequently, we decided to continue using Tukey’s HSD, as recommended by GraphPad software. To emphasize the significance of the main effects and interactions in the employed models, a Venn plot corresponding to the results has been included.

In response to your suggestion, we considered conducting inter-segmental correlation analyses between signal proteins and receptors across different sections of the GIT. However, our preliminary investigations into this approach, such as exploring the correlation between GHRL in the abomasum and GHSR in the ileum or between SMIM20 (phoenixin-14) across various GIT sections, indicated that such analyses did not yield significant new insights. Moreover, we found that incorporating these results into our study could potentially make the analysis more complex and less clear for readers. As a result, we decided not to include these extensive correlations in our discussion. We appreciate your suggestion and agree that exploring these connections could be valuable in future research to elucidate potential regulatory mechanisms.

Specific comment.

L81-L83. Can be also interesting to highlight not only the intensive growth of calves but the relevant transformation from a pre-ruminant to a ruminant condition with high implication in physiology and functions of the GIT.  

Thank you for this suggestion. As we indicated in the text, the forestomachs of the calves included in this study were already developed, as the process of stimulation with solid food had begun earlier. However, this does not change the fact that, indeed, this transformation may be reflected in the physiology and function of the GIT, as rightly was noted. We have added this information in lines 73-76.

L64-L89. A bit long this part. Probably can be summarize a little bit.

Thank you for this suggestion. This part has been summarized in the revised manuscript version.

L143. “No significant recurring ligand-receptor type correlations were identified”. Did you test suck kind of correlations between different  GIT sections? (comment above).

Thank you for this comment. As we have explained above, these types of analyses were done, but they did not contribute to any significant conclusions.

L157-159. Review writing of these two sentences. As now they seem contradictory.

Thank you for this comment. We agree, that the choice of words may have been confusing. We hope that the corrected version is more accurate (lines 178-181).

L159. Differences in reticulum between calves and adults do not seem relevant. They cluster together.

Thank you for this comment. We have corrected this information (lines 178-181).

L330-334. Four lines for discussion of correlation analysis and heatmap seems scarce. Particularly it is suggested to improve the correlation analysis approach and probably in this way the discussion (see general comment above).

Thank you for suggestion. We have improved correlation analysis description (lines 154-166 and 375-397).

L338-346. Too long paragraph?

Thank you for this comment. We have removed unnecessary information from the paragraph (lines 398-404).

L350. It can be relevant to provide information regarding the feeding regimen of these animals and production system (to evaluate p.e. the development of the rumen fermentative capacity in calves, or the possible impact of more or less forage in the diet) and also the management of the animals previous to slaughter (feed withdrawal time, transport…etc) that could determine relevant changes in the short-time metabolic control of appetite.  These elements would also deserve some discussion.

Thank you for this comment. Our primary focus was to ensure group homogeneity in our experiment, so that the only differentiating factors were age and the GIT section, while of course, the information you mentioned can be important in the perspective of the whole experiment and in terms of its reproducibility. As we mentioned before, the calves in our study were already eating solid food, so their forestomachs were fairly developed. Regarding management of the animals previous to slaughter, we have provided more details about animals' origin, nutrition, and weight in the revised manuscript in the Materials and methods section.

Round 2

Reviewer 1 Report

Comments and Suggestions for Authors

All my initial comments stand, hence, I did not reach a different conclusion on the article.

Author Response

Many thanks to the reviewer for all substantive criticisms. They have made our manuscript much better.